# Healthy Breastfeeding Infants Consume Different Quantities of Milk Fat Globule Membrane Lipids

**DOI:** 10.3390/nu13092951

**Published:** 2021-08-25

**Authors:** Alexandra D. George, Melvin C. L. Gay, Jayashree Selvalatchmanan, Federico Torta, Anne K. Bendt, Markus R. Wenk, Kevin Murray, Mary E. Wlodek, Donna T. Geddes

**Affiliations:** 1School of Molecular Sciences, The University of Western Australia, Crawley, WA 6009, Australia; melvin.gay@uwa.edu.au (M.C.L.G.); m.wlodek@unimelb.edu.au (M.E.W.); donna.geddes@uwa.edu.au (D.T.G.); 2Metabolomics Laboratory, Baker Heart and Diabetes Institute, Melbourne, VIC 3004, Australia; 3Precision Medicine Translational Research Programme and Department of Biochemistry, Yong Loo Lin School of Medicine, National University of Singapore, Singapore 119077, Singapore; jayashree.s@u.nus.edu (J.S.); bchfdtt@nus.edu.sg (F.T.); bchmrw@nus.edu.sg (M.R.W.); 4Singapore Lipidomics Incubator, Life Sciences Institute, National University of Singapore, Singapore 119077, Singapore; anne.bendt@nus.edu.sg; 5School of Population and Global Health, The University of Western Australia, Nedlands, WA 6009, Australia; kevin.murray@uwa.edu.au; 6Department of Physiology, School of Biomedical Sciences, Faculty of Medicine, Dentistry and Health Sciences, The University of Melbourne, Parkville, VIC 3010, Australia

**Keywords:** human milk, breastmilk, breastfeeding, lipidomics, lactation, fat

## Abstract

The human milk fat globule membrane (MFGM) contains important lipids for growing infants. Anthropometric measurements, milk samples, and infant milk intake were collected in a cohort of eleven healthy mother–infant dyads during exclusive breastfeeding from birth to six months. One hundred and sixty-six MFGM lipids were analysed using liquid chromatography-mass spectrometry, and the infant intake was calculated. The concentrations and intake were compared and associations between infant intake and growth characteristics explored. The lipid concentrations and infant intake varied widely between mother–infant dyads and between months one and three. The infant intake for many species displayed positive correlations with infant growth, particularly phospholipid species. The high variation in lipid intake is likely an important factor in infant growth, with strong correlations identified between the intake of many MFGM lipids and infant head circumference and weight. This study highlights the need for intake measurements and inclusion in cohort studies to elucidate the role of the human milk lipidome in infant growth and development.

## 1. Introduction

Human milk (HM) supplies infants with many nutritive and non-nutritive components, including a vast number of lipids, essential for optimal growth and development. These lipids compose approximately 5% of the total milk profile and are packaged as milk fat globules. Milk fat globules are proposed to be synthesised from triacylglycleride droplets within the endoplasmic reticulum of the mammary epithelium, and thus, the milk fat globule membrane (MFGM) is a tri-layer structure comprised of many polar lipid species surrounding a nonpolar lipid core [1]. The lipids of the MFGM make up approximately 2% of the HM lipidome and include phospholipids, such as phosphatidylinositol (PI), phosphatidylcholine (PC), phosphatidylethanolamine (PE), and phosphatidylserine (PS), and sphingolipids such as sphingomyelin (SM) and ceramides (Cer). Previous MFGM lipid research has predominantly involved basic separation techniques such as thin-layer chromatography to separate the lipid classes, and it is only recently that in-depth lipidomic analyses are being carried out [2,3,4].

The complex HM lipidome is highly variable. HM triacylglyceride concentration differences of up to 535% have been identified between women, with daily variation for some species around 100% [4,5]. Similarly, two-fold concentration differences between women have been identified for some MFGM lipid species, including phospholipids and sphingolipids [4]. Biologically, this variance can be explained in part by the maternal diet, which is known to influence the HM fatty acid composition, with previously documented correlations between the maternal fish intake and HM docosahexaenoic acid content and by other possible maternal factors, while some variation is likely due to sampling differences [6,7,8,9].

In addition to HM compositional variances, breastfeeding infants consume vastly different milk volumes, ranging from <500 to >1300 mL/day [10]. As a result, the daily intake of MFGM lipid species is expected to be an important factor in understanding the role of lipids in infant growth and development [11]. MFGM lipids are proposed to have both nutritive and non-nutritive functions for the infant, including roles in immune and neural development, and in supporting the infant gut microbiota [12]. Correlations have been identified between the HM lipid profile and preterm infant growth trajectories; however, the vast majority of MFGM lipid research fails to incorporate the infant growth, health, or developmental characteristics [3]. Understanding the quantity of lipids consumed by the infant may also be a key factor in understanding the role that they play.

In this exploratory study, we analysed the human MFGM lipids from the milk of exclusively breastfeeding women and investigated the infant intake for each MFGM lipid species and relationship between the intake and infant growth characteristics.

## 2. Methods

### 2.1. Study Design

Healthy mothers with term infants were recruited as part of a longitudinal study from birth to 6 months in Western Australia. Maternal age and infant gestational age at birth were recorded. Maternal and infant anthropometric measurements were collected monthly from birth to six months, while exclusively breastfeeding. Maternal weight (electronic scales; Seca, Chino, CA, USA) and height were measured, and the BMI (kg/m^2^) was calculated. Infant weight (electronic baby scales; Medela Inc., McHenry, IL, USA; accuracy ± 2 g), head circumference (measuring tape) and length (infantometer; Seca, Chino, CA, USA), and z-scores were calculated. At 3 months lactation, mothers were provided electronic baby scales and the infant milk intake was measured by test-weighing the infant before and after all feeds for a 24-h period. This volume was assumed to be constant from months one to six based on the existing literature [13]. In this descriptive study, samples and data from a subset of 11 mothers and infants were analysed with pre-feed HM samples from the feeding breast (<2 mL) manually expressed by the mother into sterile vials at months one and three of lactation, the first between 0600 and 0900 h and the second between 1900 and 2200 h. Informed written consent was obtained for all participants. The achievement of infant developmental milestones, according to the mothers’ health providers, were recorded. This study was approved by The University of Western Australia Human Ethics Research Office, RA/4/20/4023. Incomplete data or sample sets and maternal or infant chronic diseases were the study exclusion criteria.

### 2.2. Lipid Extraction and Quantification

The extraction solvent was prepared with methyl tert-butyl ether and methanol (7:2) and internal standards (Merck and Avanti Polar Lipids). The internal standards used were lysophosphatidylethanolamine (LPE) 14:0, lysophosphatidylcholine (LPC) 13:0, PC 13:0/13:0, PE 14:0/14:0, PS 14:0/14:0, SM 24:1, monohexosylceramide (Hex1Cer) d18:1/8:0, Cer d18:1/17:0, and PI 25:0. Ten microlitres of defrosted, homogenised HM was combined with 180 µL ammonium bicarbonate (150 mm) and 810 µL of the extraction solvent. The extraction solution was vortexed for 30 s, sonicated for 30 min, and centrifuged for 5 min (3000× *g*), and the supernatant (600 µL) was removed and dried. The final dried extract was reconstituted in 200 µL chloroform/methanol (1:1) [4].

A targeted lipidomics analysis was carried out on an Agilent UHPLC 1290 coupled to an Agilent 6495 QQQ mass spectrometer (Agilent Technologies Inc., Santa Clara, CA, USA). An injection volume of 1 µL was separated on an Agilent Zorbax Eclipse Plus C18 reversed-phase column (1.8 µm, 2.1mm × 50 mm), kept at 40 °C, with a binary solvent system of mobile phase A (40% acetonitrile in water with 10 mm ammonium formate) and mobile phase B (10% acetonitrile in isopropanol with 10 mm ammonium formate). Separation was carried out with a flow rate of 400 µL/min for a total of 15.8 min, with the following gradient: 0–2 min, 40% A; 2–14.01 min, 0% A; and 14.01–15.8 min, 80% A. Lipid species identification was carried out using dynamic multiple reaction monitoring in the positive mode. All the lipids were identified from their retention times and specific precursor and product ion transitions, carried out in MassHunter Qualitative Analysis Software B.07.00 and MassHunter Quantitative Analysis Software B.08.00 (Agilent Technologies Inc., Santa Clara, CA, USA, 2016).

Quality control (QC) dilution curves were prepared for quantitation (R^2^ > 0.99). Lipid species within the same class were normalised to their respective internal standard (excluding gangliosides, which were normalised to the hexosylceramide standard). QC was carried out analysing technical QC and pooled QC every 10 samples within the sample sequence, and individual lipids were excluded from the analyses if the QC samples (technical and/or pooled) had a coefficient of variation ≥25%. The isotopic effect of M + 3 SM m:n on PC (m−4):(n−1) was corrected for by calculating its abundance based on natural isotopic distributions and then subtracting this value from the peak area of the PC (m−4):(n−1). As the LCMS chromatographic method showed an overlap of these peaks, this correction prevented any misidentification or errors in quantification of the PC (m−4):(n−1) species. One hundred and sixty-six human MFGM lipids from ten classes: ceramide (Cer), ganglioside (GG), monohexosylceramide (Hex1Cer), dihexosylceramide (Hex2Cer), lysophosphatidylcholine (LPC), lysophosphatidylethanolamine (LPE), phosphatidylcholine (PC), phosphatidylenositine (PE), phosphatidylinositol (PI), and sphingomyelin (SM) were measured.

### 2.3. Infant Human Milk Fat Globule Membrane Lipid Intake

To estimate the infant intake for each MFGM lipid in a 24-h period, the mean of the morning and evening sample concentrations were multiplied by the measured 24-h volume intake for each infant for month one and month three.

### 2.4. Statistical Analysis

The study cohort characteristics are reported as the mean ± standard deviation, and repeated measures ANOVA was used to assess the growth characteristics from birth to 6 months (IBM SPSS Statistics for Windows, Version 26.0, Armonk, NY, USA: IBM Corp). *p* < 0.05 was considered statistically significant. The MFGM lipid concentrations were visually compared using a partial least squares-discriminant analysis (PLS-DA) constructed in Metaboanalyst 4.0 to investigate the relationships between the categorical data (i.e., time of day and month in lactation) [14]. The concentrations and infant intake for each lipid were compared using paired *t*-tests to assess if significant changes occurred between months one and three of lactation. Significance values were reported both without and with Benjamini–Hochberg correction to decrease the false discovery rate. The coefficient of variation (CV) was calculated to also compare months one and three. Pearson’s correlations were carried out to explore the possible involvement of MFGM lipid species with infant growth characteristics. This included assessing the acute influence using the month one intakes with the infant growth characteristics and the month one and month three intakes with the infant growth characteristics at month three. The longer-term influence of MFGM lipids on infant growth was assessed using the 6-month growth measurements and mean intake (mean of months one and three) of each lipid species. Where applicable, the data are presented as the mean ± standard deviation (full range).

## 3. Results

The study cohort consisted of 11 healthy mother–infant dyads. The mean maternal age at delivery was 31.7 ± 2.5 years. All the infants were born to term (mean gestational age 39.5 ± 1.0 weeks), with birth weights appropriate for the gestational age, and were exclusively breastfed during the study period. All infants grew according to their respective growth trajectories and reached the developmental milestones within the appropriate time frames. Six of the infants were female, and five were male. All mothers and infants were healthy throughout the study period (Table 1).

The 24-h milk intake was 727 ± 164 (473–894) mL/day, the average number of feeds was 11 ± 3 per day, and the average feed volume was 72 ± 20 mL/feed.

### 3.1. Human Milk Fat Globule Membrane Lipid Concentrations

A wide range of lipid concentrations for each lipid species existed between the women and between the time points. The relative lipid concentration differences were highest between months one and three (with nine PC, five SM, three Cer, four GG, five Hex1Cer, three Hex2Cer, four LPC, one LPE, and three PI species exhibiting significant differences, *p* < 0.05). In contrast, only one lipid varied significantly through the days in month one (SM 35:1), while 25 varied significantly between morning and evening in month three, predominantly the Hex1Cer and Hex2Cer species (Figure 1 and Appendix A).

### 3.2. Infant Intake of MFGM Lipids in Human Milk

The daily intake of MFGM lipids displayed wide variation between the infants in both months one and three. Similarly, the individual intakes of MFGM lipids varied between months one and three, with multiple significant differences present (*p* < 0.05), although none of the differences were statistically significant after correcting for multiple comparisons (Appendix A). The biological variation in the daily infant intake for all MFGM lipid species are displayed in Figure 2, with the PC and PE species demonstrating the largest interindividual variation in both months one and three but the highest in month three (up to 195% and 169%, respectively).

Infants consumed the PI, SM, and PC species in the highest amounts each day (Table 2 and Appendix A) and these lipids were also those that were the most variable between different infants (i.e., the mean intake that exhibited the highest standard deviation).

For 99 of the MFGM lipid species, the Pearson’s correlations demonstrated strong positive (>0.70) relationships between the intake and infant growth characteristics in months one, three, and/or six. The PC, Cer, SM, and PE species demonstrated the strongest correlations with infant weight (up to 0.77), head circumference (up to 0.95), and weight–length z-score (up to 0.80) measurements. This included the infant intake of Cer d19:1/22:0 and infant head circumference (both raw and z-score), which exhibited positive correlations in month 1 (both 0.81), in month 3 (0.72 and 0.78), and month 6 (0.90 and 0.95). Similarly, the infant weight–length z-score and intake of PI 38:5 were positively correlated in month 1 (0.74), month 3 (0.63), and month 6 (0.80). The full correlation results can be found in Appendix A.

## 4. Discussion

The concentrations and infant intake for 166 human milk fat globule membrane lipids were measured in this study, and several positive associations between the infant intake and growth characteristics were identified.

Human milk lipidomics is an emerging field, with analytical capabilities and, as a result, coverage of lipid species, improving. Existing human MFGM lipidome studies have investigated lipids as relative abundances or concentrations, with many studies analysing the total lipid classes rather than individual species. The basic separation of lipid classes was first reported using thin-layer chromatography, but more recently, LC-MS methods have achieved the separation and identification of many lipid species, including some found in very low abundance [4]. Using an LC-MS method, we achieved the quantitation of 166 MFGM lipid species. High degrees of variation both between women and between months were demonstrated, but it is important to remember that the total lipid content of human milk is dependent on breast fullness, which means that the concentration of randomly collected samples may not be biologically relevant, and thus, sampling is particularly important when linking the results to infant outcomes [11]. To ensure that concentrations between our samples were comparable, a strict sampling protocol was followed with timed morning and evening prefeed samples collected at months one and three of lactation. The compositional variation highlights the diverse and complex nature of HM and is likely important, given that significant growth differences have previously been measured between breastfed and formula-fed infants [15].

Like the concentration, the infant intake for the MFGM lipid species displayed wide variability. Previous studies have estimated the infant intake of the total PL (140 mg/day) and total SM (62 mg/day) based on full breast expression sampling and an assumed volume of 600 and 800 mL/day, respectively [16,17]. Here, we were able to calculate the accurate intake of specific lipids as we measured the milk consumed by the infant (range: 473–894 mL/day). Our analyses indicated that the overall human MFGM lipidome consumed by the exclusively breastfed infant varies from months one to three, with many considerable increases and decreases for individual lipid species. The most striking intake differences were demonstrated between the lipid intakes for different infants, including three- to five-fold differences for lipids such as PI 36:2 (range: 8.33–54.20 µmol/day), PC 36:2 (range: 4.33–37.41), SM 40:1 (range: 6.68–34.77 µmol/day), PC 34:1 (range: 1.49–21.52 µmol/day), and PI 36:1 (range: 3.08–22.76 µmol/day). These lipids were among those consumed in the highest amounts (Table 2). The wide degree of variation for each human MFGM lipid species and the overall changes from months one to three likely have biological relevance, resulting in growth and developmental differences between infants depending on their intake, which we attempted to explore. Although this study was not sufficiently powered to carry out the complex statistical confirmation of relationships between infant intake and growth characteristics, we have provided preliminary analyses and a basis for further studies to measure intakes when discerning the mechanisms through which human milk influences infant growth.

Since the plasma lipidomes of formula-fed and breastfed infants are different, and breastfed infants have lower obesity and disease risks, we hypothesized that there are MFGM lipid species that demonstrate correlations between the intake and growth characteristics at months one, three, and even following six months of exclusive breastfeeding [18,19]. In month one, the strongest correlations (Pearson’s correlation > 0.8) were present between the infant head circumference and intake of PE 35:2, Cer d18:1/25:0, PE 40:5, Cer d19:1/24:0, PC 31:0, PC-O 32:0, LPC 15:0, LPC 14:0, PC 35:2, Cer d19:1/22:0, LPC 17:1, and PE 32:1. Previous human studies have also found that MFGM-enriched infant formula resulted in better language development by age four than infants fed non-enriched formula and that infants supplemented with MFGM-enriched formula had accelerated neurodevelopmental and language skills after one year compared to those that were fed nonenriched formula [20,21]. Animal studies have also shown that rats fed with 1% complex milk lipid feed, including added PL species such as PE, increased linear growth rates and had a heightened learning capacity (spatial memory and recognition) compared to the rats on a 0.2% complex lipid feed [22,23]. This suggests that some of these human MFGM lipids and the amount in which they are consumed may be critical in modulating brain development and growth.

In month three, strong correlations (>0.7) were observed between the infant weight and infant intake of predominantly phospholipid species. Many phospholipid species may be involved in infant growth and development through choline, an acetylcholine precursor, influencing the optimal organ and brain growth (PC and LPC) through cell development and proliferation (PE) and through infant immune and vascular development (SM) [12,24]. As not all lipids in our study shared the same correlations, despite all being located on the MFGM, these data indicate that individual species may have specific roles in infant growth and development. This involvement exceeds an acute influence, and lipids may have specific roles in priming infant lipid metabolism, indicated by correlations (>0.8) between the mean intake of some phospholipid and ceramide species and the infant weight and head circumference at 6 months following exclusive breastfeeding.

All study participants were healthy, and therefore, the comprehensive collection of growth data, milk samples, and milk intake measurements from the same time points were strong in providing evidence for the importance of HM lipids. These data allowed the demonstration of the lipid profiles and infant intakes of 166 lipid species from the human MFGM at months one and three of lactation, and the investigation of the relationship to growth through six months of exclusive breastfeeding. The limiting factor in this study was the sample size of 11, which limited the statistical power to draw conclusions between the lipid species and infant growth. Furthermore, there are likely maternal factors that contribute to both the HM composition and infant health that require interrogation. These data should be considered exploratory and provide a basis for further research.

## 5. Conclusions

This exploratory study identified that human milk fat globule membrane lipids are consumed in vastly different quantities by healthy breastfeeding infants and in months one and three of lactation. The infant intake of many MFGM species positively correlates with the infant head circumference and weight throughout six months of exclusive breastfeeding, supporting the growing evidence that HM lipids contribute to infant growth and development. This descriptive study provides a basis for continued research to comprehensively understand the role that HM lipids play in infant growth and development.

## Figures and Tables

**Figure 1 nutrients-13-02951-f001:**
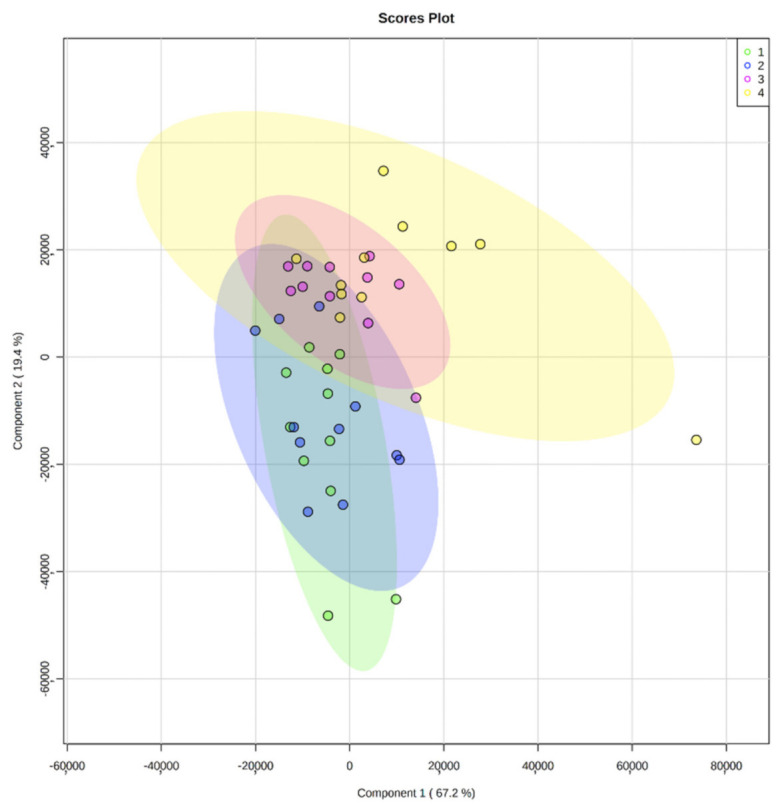
Partial least squares-discriminant analysis (PLS-DA) for human milk fat globule lipid concentrations in samples from a month one morning (green, 1), month one evening (blue, 2), month three morning (pink, 3), and month three evening (yellow, 4).

**Figure 2 nutrients-13-02951-f002:**
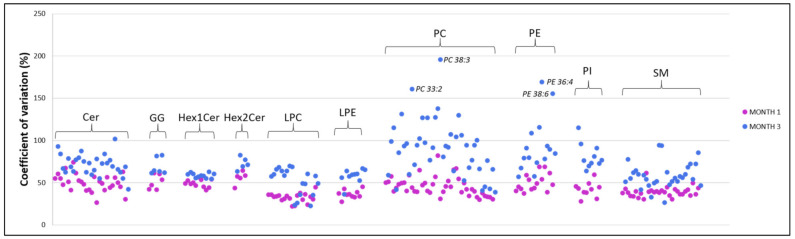
Variation (CV%) in the infant intake for each MFGM lipid species in months one and three. The species that display the highest CV (>150%) are labelled. Cer, ceramide; GG, ganglioside; Hex1Cer, monohexosylceramide; Hex2Cer, dihexosylceramide; LPC, lysophosphatidylcholine; LPE, lysophosphatidylethanolamine; PC, phosphatidylcholine; PE, phosphatidylethanolamine; PI, phosphatidylinositol; and SM, sphingomyelin.

**Table 1 nutrients-13-02951-t001:** Measured and calculated cohort characteristics from infants of birth to six months of age.

Characteristic	Delivery	Month 1	Month 2	Month 3	Month 4	Month 5	Month 6	*p*-Value
Infant weight (kg)	3.7 ± 0.4 (3.2–4.4)	4.5 ± 0.5 (4.0–5.5)	5.4 ± 0.6 (4.8–7.1)	6.1 ± 0.8 (5.3–8.3)	6.7 ± 0.9 (5.7–9.3)	7.3 ± 1.0 (6.2–10.0)	7.6 ± 1.0 (6.4–10.1)	<0.0001
Infant length (cm)	51.5 ± 2.1 (48.0–55.0)	54.8 ± 1.7 (51.9–57.4)	58.0 ± 2.1 (55.4–63.0)	60.9 ± 2.2 (58.1–64.6)	63.0 ± 2.0 (58.1–64.6)	65.0 ± 1.6 (63.2–68.3)	66.4 ± 1.9 (64.3–70.6)	<0.0001
Weight for length (z score)	0.1 ± 0.8 (−1.7–1.4)	0.2 ± 1.0 (−1.4–1.6)	0.1 ± 1.4 (−1.7–2.8)	−0.1 ± 1.1 (−1.5–2.0)	0.0 ± 1.3 (−1.4–2.9)	0.2 ± 1.3 (−1.0–3.3)	0.3 ± 1.3 (−1.4–2.9)	0.981
Infant head circumference (cm)	-	38.1 ± 0.9 (36.5–39.5)	39.8 ± 1.4 (37.5–42.0)	41.3 ± 1.2 (39.5–43.0)	42.5 ± 1.6 (40.4–45.7)	43.5 ± 1.4 (41.6–46.0)	44.5 ± 1.8 (42.0–48.0)	<0.0001
Head circumference for age (z score)	-	−0.3 ± 0.5 (−1.1–0.4)	0.9 ± 1.0 (−0.6–2.4)	0.2 ± 0.7 (−1.2–0.9)	1.2 ± 1.1 (−0.5–3.4)	1.3 ± 1.0 (−0.05–2.9)	0.9 ± 1.1 (−0.7–3.0)	<0.0001
Maternal weight (kg)	-	71.0 ± 10.8 (53.7–93.2)	70.7 ± 11.2 (51.5–93.1)	70.2 ± 12.1 (50.5–95.4)	70.2 ± 13.2 (50.0–98.3)	69.7 ± 12.1 (50.3–98.0)	70.1 ± 13.6 (50.1–97.4)	0.617
Maternal BMI (kg/m^2^)	-	25.5 ± 2.9 (21.8–31.9)	25.4 ± 3.1 (20.9–31.8)	25.2 ± 3.4 (20.5–32.6)	25.2 ± 3.7 (20.3–33.6)	25.0 ± 3.8 (20.4–33.5)	25.2 ± 3.8 (20.3–33.3)	0.479

Data are presented as the mean ± standard deviation (range). - indicates that data were not collected at these time points.

**Table 2 nutrients-13-02951-t002:** The 10 lipid species with the highest mean infant intakes (µmol/day).

Species	Intake
PI 36:2	17.15 ± 10.35
SM 40:1	13.64 ± 6.51
SM 36:1	11.05 ± 4.95
PC 36:2	10.27 ± 7.03
SM 42:2	9.71 ± 4.28
SM 34:1	8.24 ± 3.92
PI 38:4	7.21 ± 3.71
PI 36:1	7.17 ± 4.15
PI 38:3	7.10 ± 0.26
PC 34:1	6.35 ± 4.26

Data are the mean ± standard deviation.

## Data Availability

The data presented in this study are available on request from the corresponding author.

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
