# Peer review of "Healthy Breastfeeding Infants Consume Different Quantities of Milk Fat Globule Membrane Lipids"

_nutrients, 2021, doi:10.3390/nu13092951_

Round 1

Reviewer 1 Report

In breast milk research, especially in lipid-related research, it is not easy to control various variables that can affect lipid changes and collect samples on schedule. Nevertheless, I think it is a good study that was well planned from the study design to the analysis of each species, and confirmed the relationship between the results and growth.

1. However, as the authors have acknowledged, the results from 11 people seem to be difficult to conclude the relationship between some MFGM species and growth

2. Supplementary table 3 seems to be an important result related to the convention in this study. I suggest that you include it in the main text after editing the contents.

Author Response

In breast milk research, especially in lipid-related research, it is not easy to control various variables that can affect lipid changes and collect samples on schedule. Nevertheless, I think it is a good study that was well planned from the study design to the analysis of each species, and confirmed the relationship between the results and growth.

  1. However, as the authors have acknowledged, the results from 11 people seem to be difficult to conclude the relationship between some MFGM species and growth.

In response to this comment, and those from reviewer 2, the study has again been described as exploratory in the end of the discussion and in the conclusion, to ensure that it is clear that further research is required to validate and understand the data of this manuscript: “The limiting factor in this study was the sample size of 11 which limited statistical power to draw conclusions between lipid species and infant growth. Furthermore, there are likely maternal factors that contribute to both HM composition and infant health that require interrogation. The data should be considered exploratory and provide a basis for further research.”

  1. Supplementary table 3 seems to be an important result related to the convention in this study. I suggest that you include it in the main text after editing the contents.

We agree that table 3 would be nicely included as a figure within the text, however, due to the size of table 3 (at least 170 rows and 20 columns), we are not sure how Nutrients would incorporate this into the main text as either a figure or table. We would be happy to take suggestions from the editor.

Reviewer 2 Report

This exploratory study interestingly evaluates milk fat globule membrane lipid intake and infant growth in a small cohort. The unique aspects of the methods in MFGM measurement as well as the addition of infant volume intake estimates provide key insights not often accounted for in milk research and the associations of milk composition and infant growth. There are limitations in the generalizability of this study with the small sample size and concerns to be addressed as described below.

Abstract:

  • Include the duration of infant growth measurements to 6 months

Methods:

  • Ln 73: Please describe the test-weighing procedure for infants for 24 hours, were parents provided a scale to complete this at home? Is this data not available at 1 month as well to support the prior literature you sited that feeding volumes are constant from 1-6 months; other sources report increasing milk intake over the first 3-4 months of life which make your assumption of intake at 1 month less valid.
  • Study design 2.1: Include maternal and infant characteristics collected and method of collection.
  • Developmental milestone achievements are mentioned, however a description of this evaluation is not provided

Results:

  • Table 1/infant growth: Were any infants SGA or LGA at birth as this may impact their anticipated growth trajectory over the first 6 months of life?
  • Are you able to include sex specific differences in infant intake and MFGM consumption?
  • Ln 181: Consider evaluating associations with growth trajectory over the first 6 months of life in addition to individual time points as this may provide more insight into growth influences of fast/slow growth patterns.

Discussion:

  • Ln 225: Please expand upon the potential biological relevance of variation in MFGM
  • Ln 258: Include additional limitations in the confounding factors that you are not able to address in the study based on maternal health condition and maternal diet impacting milk composition.

Conclusion:

  • Please clarify that this is an exploratory study as readers interpret the generalizability and meaning of the results

Author Response

This exploratory study interestingly evaluates milk fat globule membrane lipid intake and infant growth in a small cohort. The unique aspects of the methods in MFGM measurement as well as the addition of infant volume intake estimates provide key insights not often accounted for in milk research and the associations of milk composition and infant growth. There are limitations in the generalizability of this study with the small sample size and concerns to be addressed as described below.

Abstract:

  • Include the duration of infant growth measurements to 6 months

We have added the time frame of measurements (birth to six months) to the abstract (line 16).

Methods:

  • Ln 73: Please describe the test-weighing procedure for infants for 24 hours, were parents provided a scale to complete this at home? Is this data not available at 1 month as well to support the prior literature you sited that feeding volumes are constant from 1-6 months; other sources report increasing milk intake over the first 3-4 months of life which make your assumption of intake at 1 month less valid.

We have added more specific details to line 73: “mothers were provided with electronic baby scales.”

From experience, requesting more than one 24-hour test weighing from the mothers reduces the accuracy of results and participant enthusiasm, therefore only one 24-hour test weigh was carried out. This assumption is a limitation of our study, however, the study that was cited (Mitoulas, 2002) is currently the most comprehensive study using test weighing to analyse 24 hour milk volume over lactation, describing no significant differences in milk volume from months one to six.

  • Study design 2.1: Include maternal and infant characteristics collected and method of collection.

A sentence was added to clarify that maternal age and infant gestational age at birth were recorded (line 67). A sentence was added to the study design that “Incomplete data or sample sets and maternal or infant chronic disease were study exclusion criteria,” however, there were no dyads excluded in this study.

  • Developmental milestone achievements are mentioned, however a description of this evaluation is not provided

A sentence has been added to the study design (line 80): “Achievement of infant developmental milestones, according to the mothers’ health providers were recorded.”

Results:

  • Table 1/infant growth: Were any infants SGA or LGA at birth as this may impact their anticipated growth trajectory over the first 6 months of life? All infants in the study were born term with weights that were appropriate for gestational age. This has been added to the results section (line 145).
  • Are you able to include sex specific differences in infant intake and MFGM consumption?

Due to the small sample size we were cautious to suggest potential sex specific differences. To overcome this, growth z-scores were also used for analysis, to ensure growth differences between male and female infants were considered. 

  • Ln 181: Consider evaluating associations with growth trajectory over the first 6 months of life in addition to individual time points as this may provide more insight into growth influences of fast/slow growth patterns.

We agree that this would be a worthwhile analysis. However, similar to above, the limited sample size meant that statistical analysis of lipid species and different growth trajectories would be far too weak.

Discussion:

  • Ln 225: Please expand upon the potential biological relevance of variation in MFGM

This sentence has been added to in order to clarify that the variation that we measured likely has biological relevance: “The wide degree of variation for each human MFGM lipid species, and the overall changes from months one to three likely have biological relevance, resulting in growth and developmental differences between infants depending on intake, which we attempted to explore.”

  • Ln 258: Include additional limitations in the confounding factors that you are not able to address in the study based on maternal health condition and maternal diet impacting milk composition.

As the focus of this study was on the human milk lipid species that may be influencing infant growth, maternal factors that may affect milk composition or infant growth were not deeply interrogated. This section has been expanded to clarify: “The limiting factor in this study was the sample size of 11 which limited statistical power to draw conclusions between lipid species and infant growth. Furthermore, there are likely maternal factors that contribute to both HM composition and infant health that require interrogation. The data should be considered exploratory and provide a basis for further research.”

Conclusion:

  • Please clarify that this is an exploratory study as readers interpret the generalizability and meaning of the results

In response to this comment, and those from reviewer 1, the study has again been described as exploratory in the end of the discussion and in the conclusion, to ensure that it is clear that further research is required to validate and understand the data of this manuscript: “The limiting factor in this study was the sample size of 11 which limited statistical power to draw conclusions between lipid species and infant growth. Furthermore, there are likely maternal factors that contribute to both HM composition and infant health that require interrogation. The data should be considered exploratory and provide a basis for further research.”

The study is also described as exploratory and limited by sample size in the introduction (line 61) and in the discussion (line 230) in order to remind the reader that these results require follow-up.